# Pediatric Population with Down Syndrome: Obesity and the Risk of Cardiovascular Disease and Their Assessment Using Omics Techniques—Review

**DOI:** 10.3390/biomedicines10123219

**Published:** 2022-12-12

**Authors:** Marta Hetman, Ewa Barg

**Affiliations:** Department of Basic Medical Sciences, Wroclaw Medical University, 50-556 Wroclaw, Poland

**Keywords:** Down syndrome, obesity, metabolomics, lipidomics, cardiovascular disease

## Abstract

People with Down syndrome (PWDS) are more at risk for developing obesity, oxidative stress disorders, metabolic disorders, and lipid and carbohydrate profile disorders than the general population. The presence of an additional copy of genes on chromosome 21 (i.e., the superoxide dismutase 1 gene (SOD1) and gene coding for the cystathionine β-synthase (CBS) enzyme) raises the risk for cardiovascular disease (CVD). As a result of disorders in metabolic processes and biochemical pathways, theoretically protective factors (low homocysteine level, high SOD1 level) do not fulfil their original functions. Overexpression of the CBS gene leads to the accumulation of homocysteine—a CVD risk factor. An excessive amount of protective SOD1, in the case of a lack of compensatory increase in the activity of catalase and peroxidase, leads to intensifying free radical processes. The occurrence of metabolic disorders and the amplified effect of oxidative stress carries higher risk of exposure of people with DS to CVD. At present, classic predispositions are known, but it is necessary to identify early risk factors in order to be able to employ CVD and obesity prophylaxis. Detailed determination of the metabolic and lipid profile may provide insight into the molecular mechanisms underlying CVD.

## 1. Introduction

Diagnosing a disease before its first symptoms appear poses a great challenge in medicine. In the past, it was believed that only genes affected the maintenance of homeostasis and health in the body. After many years of research and genome sequencing, we are still unable to diagnose numerous diseases or to design an effective therapy at an early stage. It was assumed that the organism is a complex structure, at the base of which is the genome, and its subsequent levels being transcripts, proteome and metabolome [1]. Metabolism plays a key role in all areas of biology, which is why more and more of those areas are being studied from its perspective. Many adult conditions, such as cardiovascular and metabolic disease, originate during childhood, therefore the knowledge of risk factors for these diseases will allow for the implementation of an early and effective prophylaxis.

Down syndrome (DS, trisomy 21, T21) is the most common chromosome abnormality (caused by trisomy of the whole or a part of chromosome 21) with a worldwide incidence rate of 1:1000–1100 in newborns [2]. The extra chromosome 21, or at least a portion of it, results in a constellation of clinical features (cardiac defects, delayed growth, hematology and endocrine abnormalities, autoimmune diseases, intestinal, stomatognathic disturbances, vision and hearing defects and obstructive sleep apnea, and others) [3,4]. Additionally, people with DS (PWDS) are at increased risk for cardiovascular diseases (CVD) (mitral valve prolapse, endocarditis, atherosclerosis (AS) and congestive heart failure [5]), pulmonary hypoplasia, muscle hypotonia, osteoporosis, arthritis, osteoarthritis, and diabetes mellitus [6,7]. There is no specific DS phenotype. Individuals may differ from each other both in terms of external features and chronic diseases. Such a complex condition contributes to the demand for profound medical care. Advancements in medicine have led to a marked improvement in life expectancy of PWDS, with the estimated median age of survival approaching 60 years [8]. Obesity-related diseases (CVD, cancer, type II diabetes, and others) have received more attention [9]. The problem of body weight disorders among PWDS is complex and challenging, concerning mainly the rapid transformation between undernutrition in the first period of life and excessive weight gain in later years. For the purposes of this review, however, we will focus on the problems of excess body weight in children and adolescents with DS. The appearance of new disease entities in this population is a challenge for health practitioners. Due to the burden of many conditions, PWDS should be monitored from an early age with the constraints associated with their health status. 

## 2. Omics Techniques: Metabolomics and Lipidomics

Omics techniques are a rapidly evolving field of molecular sciences. Metabolomics, a relatively young branch of omics science, is an interdisciplinary approach that encompasses biology, chemistry, and bioinformatics. The metabolomic examination should enable the detection of abnormalities in the patient’s health at an early stage of the development of clinical symptoms or even before their manifestation [10]. The advantage of metabolomics research is its low invasiveness, thanks to the use of mainly readily available body fluids, i.e., blood serum, plasma, saliva, urine, and tissues after prior preparation. It provides a modern bioanalytical tool to define perturbations in metabolic pathways and enables the detection of predictive biomarkers. Thus, metabolomics can contribute to more efficient diagnosis, treatment, and disease prevention. The definition of metabolomics officially appears in the literature in 1999 as “the quantitative measurement of the dynamic multiparametric metabolic response of living systems to pathophysiological stimuli or genetic modification” [11]. Metabolomics is based on the qualitative and quantitative study of small-molecule (<1.5 kDa) compounds that are intermediates and products of metabolism (lipids, amino acids, short peptides, nucleic acids, sugars, alcohols, or organic acids) and reflects in endogenous metabolism and exogenous sources such as, among others, diet or physical activity. Metabolites are involved in all biochemical reactions (any process that occurs in the body is reflected in the metabolome) and measuring them can potentially evaluate the state of the organism. There are two approaches in metabolomics: untargeted and targeted. The untargeted approach makes it possible to identify metabolic new biomarkers; the targeted approach identifies and quantifies a limited number of known metabolites. Both the presence and absence of specific metabolites can be the source of information about possible disorders in the patient’s health and draw attention to a medical problem.

Lipid homeostasis is essential for maintaining full health; therefore its evaluation is of fundamental importance. Any abnormalities in lipid metabolism play an important role in many diseases, including metabolic syndrome, diabetes, CVD, lipodystrophies, neurological/neurodegenerative disorders, and central nervous system damage [12]. Most cases of CVD are difficult to associate with well-known risk factors. Many patients, despite having optimal blood lipid levels, are exposed to CVD [13]. Therefore, it is important to search for new biomarkers that will enable the very early diagnosis and effective prevention of CVD. Single biomarkers in cardiology are very effective in confirming the occurrence of an acute event. However, it is very difficult to precisely estimate the risk of atherosclerotic disease at an early stage. So far, researchers have relied on well-established risk factors, such as smoking, hypertension, dyslipidemia, and diabetes as risk factors of developing CVD [12,14,15,16]. Detailed determination of the metabolic profile may provide insight into the molecular mechanisms underlying AS [13,17,18,19,20]. 

### 2.1. Metabolomics 

Metabolomics is a very promising tool for investigating human health, however, the analysis of the metabolome is challenging for many reasons, among others, different analytic approaches and the lack of standardization. The techniques used in metabolomics are magnetic resonance (NMR) spectrometers, mass spectrometers (MS), gas-chromatography (GC), liquid chromatography (LC) systems, ion mobility systems (IMS), capillary electrophoresis (CE) systems, integrated liquid chromatography-mass spectrometry (LC-MS), integrated capillary electrophoresis-mass spectrometry (CE-MS), integrated ion mobility spectrometry-mass spectrometry (IMS-MS) and gas chromatography-mass spectrometry (GC-MS) [21]. 

As a novel technique, metabolomics can provide insight into obesity and the risk of cardio-metabolic complications, and be used to uncover pathways underlying diet–disease associations. The heart, being a metabolically active organ [22], and its diseases constitute one of the main targets of the omics’ techniques of today. The results of metabolomics studies help to clarify the pathophysiology of many diseases, optimize treatment, and distinguish specific diagnostic biomarkers, which is especially important in the case of asymptomatic diseases [23]. In 1991, the first biomarkers of coronary heart disease using NMR spectroscopy [24] were discovered. The introduction of metabolomics to epidemiological research is of great importance for the understanding of pathophysiology and the discovery of new biomarkers for the early prevention and detection of AS and CVD [25]. In 2017, the American Heart Association published a statement on potential health and CVD effects of metabolomics and its current challenges in clinical practice [26].

Understanding the pathogenesis of childhood obesity with the help of molecular studies is one of the major challenges of current medicine. Metabolomic studies on obesity and comorbidities are conducted in adults on a large scale. Unfortunately, little research has been completed on groups of children and adolescents [27]. Hellmuth et al. combined metabolomics data from four large European cohorts finding a strong positive association of sphingomyelin (SM) 32:2 (molecular species containing myristic acid and sphingadienine) and lyso-phosphatidylcholine (LPC) 14:0 with BMI z-score (this metabolite was found to have a positive association with BMI adults [28]) and no association of non-esterified fatty acid (NEFA) 16:1 with BMI z-score [29]. An LPC 14: 0 was considered a predictor of obesity at 6 years of age (study of serum in 6-month-old infants) [30]. The rate of FA 14:0 was also elevated in phospholipids (study among obese 15-year-old children) [31]. The authors [29] additionally concluded that the concentration of lipids with 14:0 (exception of NEFA 14:0) is seemingly higher in children with high BMI and may subsequently be used more often for the synthesis of SM. In addition, the 14:0 synthesis can be enhanced by a high-calorie diet and high glycemic load of food. SM 32:2 may be a potential biochemical marker for the combined effect of genetic predisposition, high dietary intake of total energy, glycemic load, and linoleic acid [29]. Atherosclerosis, the major cause of CVD, is often attributed to lifestyle factors [32]. A high risk of the early development of AS has been proved in people with hyperhomocysteinemia, hypermethioninemia, and homocystinuria [33]. Wurtz et al. identified phenylalanine and various fatty acids as biomarkers for CVD [34]. Biomarkers related to insulin resistance and energy metabolism have also already been identified [35]. A consistent metabolic profile of childhood obesity was observed including amino acids (particularly branched chain and aromatic), carnitines, lipids, and steroids [36,37].

### 2.2. Lipidomics

Lipidomics, a discipline belonging to metabolomics, is described as the quantitative characteristic of the complete lipid complex [17]. The subject of research in this subdiscipline is lipids, i.e., a functional unit characterizing the molecular lipid image of a biological sample under study. However, thanks to lipidomics, it is possible to quantify various lipid molecules (acylglycerols, sterols, sphingolipids, and others) [17,19]. Lipidomic evaluation allows for a picture of lipid concentrations, for example, the total plasma lipidomics of the tested total plasma shows a detailed and much more complete picture of lipid metabolism and possible abnormalities of lipid metabolism—as opposed to studies of isolated lipoproteins [12]. Lipidomics had identified ceramides and sphingolipids as potential mediators of cellular dysfunction.

Lipidomics study the structural, signaling and metabolic functions of lipid compounds. Due to the large variety of lipid types in cells/tissues, the lack of homogeneity and their frequent biochemical modifications, detailed characterization may be a difficult task [38]. For these reasons, data on lipidomics rarely appear in the scientific literature when compared with other “omics” technologies. Thus, a sufficient amount of data on the relationship between lipid metabolites with CVD are still lacking [39]. Lipids are presented as CVD risk factors only in major classes and not as individual molecular entities in diabetes [40].

The main challenge of lipidomics is the demonstration of new risk factors and the early detection of the risk of atherogenesis at the clinical level. It seems necessary to identify early risk factors in order to undertake CVD prophylaxis. 

## 3. Down Syndrome 

### 3.1. Cardiovascular Disease

In 1977, Murdoch et al. found a complete absence of atherosclerosis in five posthumously examined PWDS [41]. In addition, Pueschel et al. confirmed the lack of significant differences in the level of total cholesterol (TC), low-density lipoprotein (LDL), apolipoprotein B (apoB), and apoB/apolipoprotein A (apoA) between the examined group of PWDS and the control group [42]. As a result of conducted research in the 1970′s, it was thought that people with DS were no more at risk of atherosclerosis than the general population. However, the lifestyle and eating habits differed significantly from those present, and life expectancy was much shorter. Recently, there have been scientific studies that may suggest that significant and known risk factors for CVD and AS have been observed among people with DS: diabetes [43,44], obesity [43,45,46,47] and hypertension [44], and lipid disorders [48,49]. At the same time, it has been shown that PWDS have a lower incidence of AS [41,44,50,51,52,53,54,55,56]. Additionally, lower BPs at rest may have a protective role against the development of atherosclerosis in PWDS [53,57]. 

Interestingly, Landes et al. showed that PWDS were more likely to die at younger ages from heart diseases compared with the general population [58]. The study of Hill et al., Day et al., and Hermon et al. showed an increased risk of death for PWDS due to CVD in comparison with the general population [59,60,61]. However, Torr et al. analyzed morbidity and mortality among PWDS and indicated ischemic heart disease to be a minor cause of death [62]. Adelekan et al. found that children with DS have less favorable lipid profiles than their siblings [8]. Sheela et al. showed that youth with DS had more atherogenic lipid and lipoprotein particle profiles, including higher LDL-C levels, compared with those without DS [49]. Buonouomo et al. found high levels of TC, LDL-C, and TG and low HDL-C in individuals aged 2–9 years old with DS [62]. This study group with DS also had a higher prevalence of prediabetes and an increased amount of visceral fat [49]. In general, the increased LDL-C level in youth with DS reveals a greater risk of atherosclerosis. Adults with DS also have a high risk of stroke, driven largely by high cardioembolic risk [44]. 

Lipoprotein(a) (Lp(a)) seems to be involved in the pathogenesis of CVD [63]. Krzesińska et al. compared lipid parameters, protein composition, antioxidative properties of HDL, and Lp(a) levels in adolescents with DS and healthy individuals [64], and showed unfavorable lipid profiles in conjunction with significantly higher Lp(a) levels and quality changes in HDL particles in adolescents with DS. Serum Lp(a) levels are relatively stable over a lifetime [65], therefore a once-in-a-lifetime Lp(a) measurement could help identify those at increased risk of CVD [66]. Data appearing in the literature seem to be contradictory. Most, however, argue for the need to refute the belief that DS is a disease free from atherosclerosis. In this situation, it is advisable to extend the research on atherosclerosis risk factors and predisposition to related diseases in people with DS with the use of omics techniques. 

### 3.2. Excessive Body Weight and Physical Activity

The literature repeatedly reports that DS children are predisposed to obesity [67,68,69,70,71,72], abnormal or excessive fat accumulation caused by a positive energy balance, which has been associated with a negative impact on health [73]. Adults with DS are twice as likely to be obese and nearly four times more likely to be extremely obese in comparison with adults without DS [74]. The literature describes potential causes of obesity tendency among children and adolescents with DS: decreased energy expenditure at rest; increased leptin levels; untreated hypothyroidism; unhealthy diet; and low physical activity [70,75]. Additionally, children and adolescents with DS show less physical activity than their peers without DS [76,77,78], although the tested level of physical activity in adolescents without DS turned out to be insufficient in 80% of them [79]. What is worse, PWDS tend to become less active as they become older, with higher rates of obesity in girls [80,81,82,83,84]. Although the activity level among children with DS was lower, the caloric intake was higher in this group [75]. The greatest acceleration in obesity occurs between the ages of 2–6 years [84]. In the teenage period, when PWDS gain more independence and the ability to choose the type and amount of food (with the predominance of processed products with excessive amount of salt and sugar), obesity begins to be most visible. Yahia et al. pointed out that prepubertal obese-DS displayed excess body adiposity with pronounced central fat distribution, atherogenic lipid profile, and higher insulin resistance compared with matched obese-control [85,86]. Wernio et al. pointed out that overweight children with DS were characterized by higher levels of triglycerides, atherogenic index of plasma, and apoA2 and apoE levels [87]. Obesity also contributes to the worsening of obstructive sleep apnea symptoms and the burden of congenital heart disease [70,88,89]. With age, it becomes more and more difficult to persuade teenagers to play sports regularly. However, the DONUT STUDY showed that, despite potential difficulties in the pursuit of a correct diet and inadequate approach to physical activity, children with DS could achieve results that are substantially the same as those of non-DS children [90]. Moreover, some children and adolescents with DS are limited by reduced respiratory efficiency and congenital heart diseases [81]. An additional obstacle to increasing physical activity among PWDS is the COVID-19 epidemic that has been present for over 2 years. Amatori et al. showed a negative impact of COVID-19: decreased physical activity and increased sedentary behaviors [91]. It is worth remembering that the patterns of proper nutrition should function throughout a household. Stefanowicz-Bielska et al. proved that in families of overweight and obese children with DS, other members had nutritional disorders more frequently [92]. Caregivers and siblings should be equally involved in shaping healthy habits and lifestyle. Different levels of intellectual disability can also make it difficult to make correct food choices. Hence the repeated emphasis on the importance of the role of the family as a promoter of a healthy lifestyle. Roccatello et al. analyzed meals of choice of the people with DS finding bread, pasta and sweets as their favorite go-to foods [92]. The least-liked food was vegetables. Fruit juices and ready-to-drink tea were the main sources of simple sugars [92], which can contribute to liver steatosis and hypertension (the impact of fructose) [75]. Introducing healthy eating habits may be fundamental to sustaining good health. Jobling et al. conducted an intervention study (education program) [93]. The program was successful in convincing people with DS to reduce their consumption of sweets but the researchers’ actions did not change other unhealthy eating habits [93]. However, Naczk et al. enrolled adolescents with DS in a thirty-three weeks swimming program that resulted in decreases in body mass, body fat, and BMI [94]. Because regular physical activity is recommended to reduce the risk of developing health conditions such as heart disease, cancer, type 2 diabetes, high blood pressure, osteoporosis, and obesity [95,96], sports programs of this type play a very important role in acquired heart-disease prevention. As children and adolescents with DS are predisposed to overweight and obesity, and also tend to be physically inactive, they are at a significant risk of mortality and many serious diseases. Me et al. have shown that breastfeeding may be a protective factor for obesity and high body fat in children [97]. In 2022, a systematic review of DS and breastfeeding was conducted: around 50–23.3% of the children with DS were never breastfed and rates of breastfeeding in infants with DS were lower than those in controls in three studies [98]. 

### 3.3. Oxidative Stress

PWDS have been identified as having high oxidative stress (the imbalance between free radical production and the prooxidative state within the cell determines a biological state [99]), which is connected with the risk of the development of AS, neurodegeneration, cell ageing, cancer, and immunological disorders [100,101]. Oxidative stress, which damages blood vessel tissues, also plays a role in the pathogenesis of AS. In oxidatively damaged tissues, the development of AS is facilitated [102]. Endothelial cell function may be impaired in PWDS despite their protection against AS [103]. Furthermore, high oxidative stress has been related to elevated insulin resistance, poor insulin sensitivity, and hypertension [104]. It has been shown that T21 is associated with pro-oxidant status and increased susceptibility to oxidative damage [105,106,107]. T21 of the chromosome increases the representation and expression of Cu/Zn superoxide dismutase (SOD1), the gene which is located on the distal segment of chromosome 21 (21q22.1) [108]. It has been shown that PWDS have an increased SOD1 activity by as much as 150% compared with people without DS [109]. SOD-1 is the main enzyme in the antioxidant defense system. Under physiological conditions SOD-1, together with catalase and peroxidases, protects the body against the harmful effects of very reactive free oxygen radicals which are a potential threat to cellular structures. Radicals and reactive oxygen species (ROS) are formed during normal cellular metabolism, however, in the conditions of their increased formation or disturbances in the activity of antioxidant enzymes, free radical damage occurs. It is believed that excessive SOD-1 activity is responsible for the increased formation of hydrogen peroxide and it heightens the risk of oxidative stress (prolonged higher SOD activity may lead to glutathione depletion, deficiencies in catalase and peroxidases’ activity). SOD-1 catalyzes the conversion of the peroxide anion to hydrogen peroxide, which leads to the continuous production of two major reactive oxygen species in oxygen cells in the mitochondria [109]. In the pathogenesis of AS, ROS are responsible for the formation of oxidatively modified LDLs (oxyLDL), which are pro-atherogenic substances [110,111,112]. The biological effects of ROS are controlled by a wide spectrum of antioxidant defense mechanisms such as the action of vitamins E and C, uric acid, glutathione, and antioxidant enzymes. The reduced concentration of glutathione in the blood along with the overexpression of the SOD-1 gene in PWDS additionally contributes to elevated exposure to the negative effects of oxidative stress. The increased activity of SOD-1 as an antioxidant enzyme could explain the protective role in preventing atherosclerotic lesions. In the situation of disturbed antioxidant balance, as is the case in DS, due to the lack of compensatory higher activity of catalase and peroxidase, free radical processes are intensified. It is known that high SOD-1 activity means a disturbed balance of the antioxidant system: the peroxidation processes of lipid peroxides, participating in the formation of atherosclerotic plaques, dominate. In people who experience increased oxidative stress, biologically important molecules such as lipids, proteins, or nucleic acids are oxidized, which significantly affects the incorrect function of both individual organs and the entire body. Therefore, it seems that PWDS will be additionally exposed [111]. Lipid peroxidation (LPO) in free radical reactions is the process of oxidation of unsaturated fatty acids or lipids in which peroxides of these compounds are formed. They are an important link in the atherosclerotic process [112]. They modify physical properties of cell membranes and inhibit the activity of membrane enzymes and transport proteins. There is also a link between an aerobic modification of LDL-C and inflammatory activity of macrophages through the induction of macrophage cyclooxygenase 2 expression by LPO products [110]. Chronic oxidative stress leads to the intensification of degenerative processes and premature aging of tissues. With age, numerous hormonal and metabolic disorders appear and worsen, which is the leading problem in older children and adolescents with DS who require constant and targeted medical care. An earlier description of DS as a “non-atherosclerotic model” could be justified, inter alia, by increased activity of the defense enzyme SOD-1 and the altered metabolism of homocysteine. The results of the current research seem to contradict this assumption. In this situation, it is advisable to extend the research workshop on atherosclerosis risk factors and predisposition to other diseases in people with DS to include metabolomics research. Wernio et al. pointed out that fat mass, fat mass/height2 index, and visceral fat mass in children with DS correlated with advanced oxidative protein product level [87]. Figure 1 shows the processes described above.

### 3.4. Metabolic and Endocrinological Disorders 

The gene coding for the enzyme β-cystathionine synthase (CBS) is located on chromosome 21. This enzyme is responsible for converting homocysteine (Hcy) and serine into cystathionine in the methionine metabolic pathway. Hcy, being a by-product of methionine metabolism, must be converted back to methionine (by re-methylation or by conversion to cysteine). In this process, the important part is played by folic acid and vitamins B (B6 and B12). Three copies of the CBS enzyme genes cause its overexpression; thus, people with DS have a reduced level of homocysteine, which should result in a reduced risk of AS. The reduced concentration of homocysteine also means a lower concentration of methionine, deficiency of tetrahydro folic acid (THF) (the so-called THF trap), and the participation of B vitamins in the methionine pathways. Additionally, the low concentration of homocysteine results in DNA hypermethylation. The disruption of the methionine metabolism pathways is caused by a number of unfavorable metabolic disorders that can be detected using metabolomics studies. Low availability of vitamin B (B6, B12, and folic acid) leads to impaired re-methylation of homocysteine to methionine and thus accumulation of homocysteine [113]. Recent data indicate that homocysteine accumulates in states of increased oxidative stress associated with immune activation [113]. To better understand the above processes, simplified diagrams have been prepared: Figure 2 shows the correct metabolic pathways; Figure 3 shows the disorders occurring in DS.

Children with DS have a higher likelihood of developing endocrine and metabolic disorders such as thyroid dysfunction, diabetes mellitus, short stature, vitamin D deficiency, and obesity than the general population [44,89,114,115,116,117,118]. Thyroid dysfunction is the most common endocrine abnormality in DS children: it is about 38 times more common in individuals with DS than in other people [119,120]. Thyroid hormones are involved in the regulation of carbo–lipid metabolism. They are related to oxidative stress by stimulating cellular metabolism and influencing antioxidant mechanisms as well as regulating oxygen consumption and producing free radicals [111,121]. It is estimated that the incidence of thyroid gland disorders in people with DS increases with age [122]. Aslam et al. demonstrated that at younger ages the incidence of diabetes in patients with DS is four times higher than that of control patients. Peak mean BMI is higher and established earlier in DS, contributing to T2DM risk [123]. The prevalence of type 2 diabetes mellitus in children with DS ranged between 0% and 3.6% [117]. Wernio et al. pointed out that in children with DS fat mass, fat mass/height^2^ index, and visceral fat mass correlated with thiobarbituric acid reactive-substances and advanced oxidative protein product-levels [87]. 

### 3.5. Metabolomics in Down Syndrome

To date, numerous disturbances in the concentration of metabolites in DS have been described, such as: increased levels of phenylalanine and tyrosine in blood serum [124]; lower plasma levels of free histidine, lysine, tyrosine, phenylalanine, leucine, isoleucine, and tryptophan [125]; increased plasma concentrations of leucine, isoleucine, cysteine, and phenylalanine [126]; decreased plasma concentration of serine [127]; increased plasma lysine concentration [127]; elevated concentrations of metabolites related to the methylation cycle such as cysteine, cystathionine, choline, and dimethylglycine [125]; and increased concentrations of S-adenosylhomocysteine and S-adenosylmethionine [125]. Little data exist on the use of metabolomics among PWZD [125,126]. At the same time, there are no data on the use of lipidomics in DS. Orozco et al. analyzed metabolomics of 31 PWZD and observed alterations to methylation metabolism, carnitine/O-acetyl carnitine, dimethyl sulfone, and myo-inositol in children with DS [128]. Obeid et al. reported similar findings in methylation pathway metabolites and found elevated blood cystathionine, cysteine, betaine, choline, and N,N-dimethylglycine in children and young adults with DS [125]. Caracausi et al. analyzed plasma and urine of children with DS and revealed DS/normal ratio in plasma being 1.23 (pyruvate), 1.47 (succinate), 1.39 (fumarate), 1.33 (lactate), and 1.4 (formate) [126]. As most of the altered concentrations were consistent with the 3:2 gene dosage model, there is a possibility that the mentioned changes are caused by the presence of three copies of chromosome 21 [126]. As a result of the use of different methods of omics techniques, as well as the differences in metabolites among children and adults, it is very difficult at the present stage to compare the results obtained in the studies mentioned in the review. However, it is very important to perform metabolomic and lipidomic tests in children with DS in order to be able to compare and analyze the data.

## 4. Summary and Conclusions

Trisomy of 21 chromosome affects the cardiovascular system in anatomical and physiological ways. Numerous hormonal and metabolic disorders are a leading problem in children and adolescents with DS. Those disorders aggravate with age and require constant targeted medical care. As a result of disorders in metabolic processes and biochemical pathways, theoretically protective factors (low homocysteine level, high SOD1 level) do not fulfil their original functions. The results of the current research seem to contradict the assumption that PWDS are not at risk of developing cardiovascular disease. At present, some classic predispositions are known but CVD prophylaxis requires identifying early risk factors. In such case, it is advisable to extend the research of omics techniques on atherosclerosis risk factors and predisposition to include related diseases in people with DS.

## Figures and Tables

**Figure 1 biomedicines-10-03219-f001:**
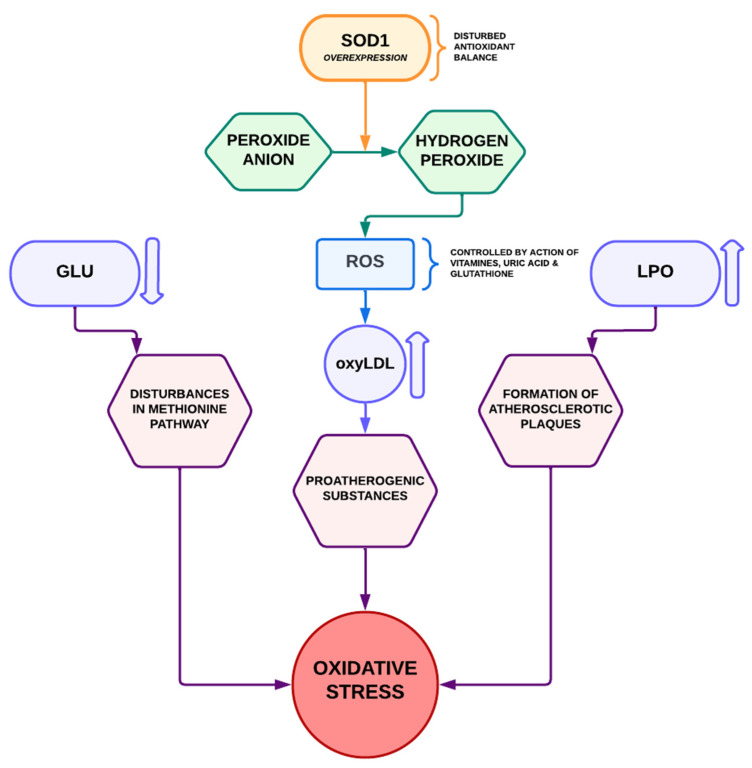
Oxidative stress processes in people with Down syndrome. SOD1-Superoxide dismutase; GLU-glutathione; ROS-reactive oxygen species; LPO-lipid peroxidation; oxyLDL-oxidatively modified LDLs.

**Figure 2 biomedicines-10-03219-f002:**
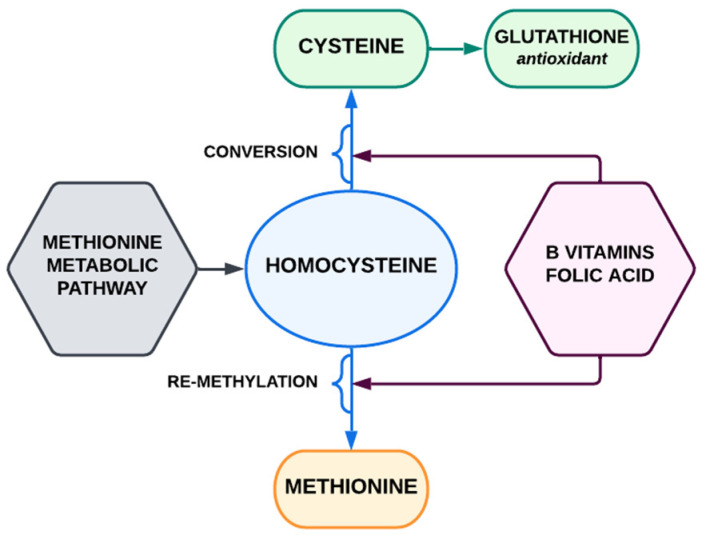
Methionine metabolic pathway (simplified).

**Figure 3 biomedicines-10-03219-f003:**
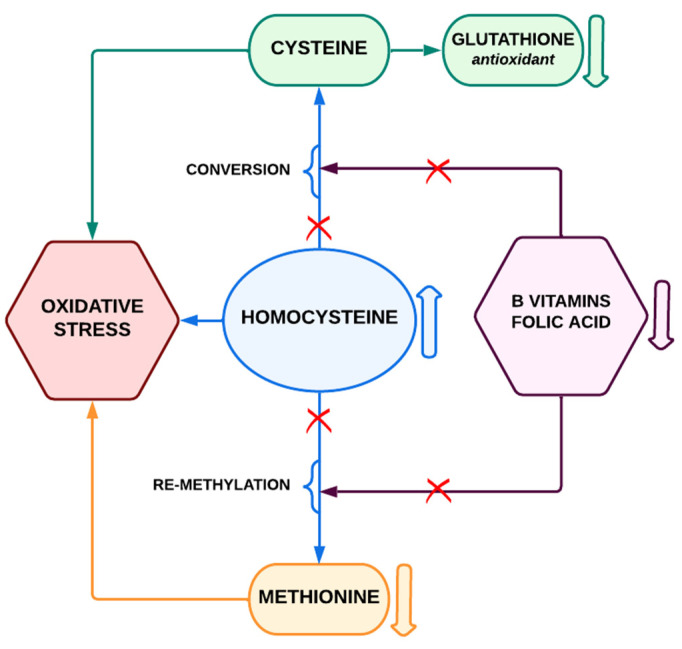
Methionine metabolic pathway (simplified)–pathway disorders in people with Down syndrome.

## Data Availability

Not applicable.

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
