# Peer review of "Pediatric Population with Down Syndrome: Obesity and the Risk of Cardiovascular Disease and Their Assessment Using Omics Techniques—Review"

_biomedicines, 2022, doi:10.3390/biomedicines10123219_

Round 1
Reviewer 1 Report
Hetman and Barg propose the review article entitled "Pediatric population with Down syndrome: obesity and the risk of cardiovascular disease and their assessment using omics techniques. Review. " In this interesting review, they look at metabolomics in Down syndrome.
Major
Figures 1 and 2 appear after figure 3. Figures in a whole are very standartized and not really eye-catching for the reader.
Introduction.
Concerning the sentence "In the past, it was believed that only genes were responsible for maintaining balance and health in the body." I believe it is innacurate. It si the regulation of the Genes, and its deregulation, that is responsible for disease, not the gene itself.
Minor
abstract : repetition of INCREASE in "The occurrence of metabolic disorders and the increased effect of oxidative stress increase the exposure of people with DS to CVD."
Introduction : please rewrite "It has allowed noticing the occurrence of obesity..."
Summary please rewrite "an assumption that an assumption that DS is free-atherosclerotic syndrome, however, cannot be clearly stated.". The sentence can be misleading. What do the authors mean by " DS is free-atherosclerotic syndrome,"
Author Response
We greatly appreciate the thorough and thoughtful comments provided on our submitted article. We have revised our manuscript, according to the reviewers’ comments, questions, and suggestions. Regardless of the reviewers' suggestions, linguistic corrections were introduced, both in terms of vocabulary and grammar. We believe that the manuscript has been further improved.
Major:
- Figures 1 and 2 appear after figure 3. Figures in a whole are very standartized and not really eye-catching for the reader.
Thank you for pointing this out. The correction has been made. Changes have also been made to the graphics of figures.
- Concerning the sentence "In the past, it was believed that only genes were responsible for maintaining balance and health in the body." I believe it is innacurate. It si the regulation of the Genes, and its deregulation, that is responsible for disease, not the gene itself.
Thank you, you have raised an important point here. We decided to erase this sentence.
Minor
- Abstract : repetition of INCREASE in "The occurrence of metabolic disorders and the increased effect of oxidative stress increase the exposure of people with DS to CVD.
Thank you for pointing this out. The correction has been made.
- Introduction : please rewrite "It has allowed noticing the occurrence of obesity..."
Thank you for pointing this out. The correction has been made.
- Summary please rewrite "an assumption that an assumption that DS is free-atherosclerotic syndrome, however, cannot be clearly stated.". The sentence can be misleading. What do the authors mean by " DS is free-atherosclerotic syndrome,"
Thank you for this suggestion. . To avoid confusion - we rewrote the sentence. This phrase was taken from the scientific work entitled Down's syndrome: an atheroma-free model? (Murdoch et al.).

Reviewer 2 Report
The authors conducted a detailed study of cardiac and obesity risk factors, lipid peroxidation, and oxidative stress in Down syndrome.
the strength of the article
The authors scanned the literature in detail and produced a good article.
Limitations of the article
The article has no explicit limitations.
Author Response
We greatly appreciate the thorough and thoughtful comments provided on our submitted article. We have revised our manuscript, according to the reviewers’ comments, questions, and suggestions. Regardless of the reviewers' suggestions, linguistic corrections were introduced, both in terms of vocabulary and grammar. We believe that the manuscript has been further improved.
